# Tau PET correlates with different Alzheimer's disease-related features compared to CSF and plasma p-tau biomarkers

Rik Ossenkoppele[1,2,*] (iD), Juhan Reimand[2,3,4], Ruben Smith[1,5], Antoine Leuzy[1], Olof Strandberg[1], Sebastian Palmqvist[1,6] (iD), Erik Stomrud[1,6], Henrik Zetterberg[7,8,9,10], the Alzheimer's Disease Neuroimaging Initiative[†], Philip Scheltens[2] (iD), Jeffrey L Dage[11] (iD), Femke Bouwman[2], Kaj Blennow[7,8] (iD), Niklas Mattsson-Carlgren[1,5,12] (iD), Shorena Janelidze[1] & Oskar Hansson[1,6,**] (iD)

## Abstract

PET, CSF and plasma biomarkers of tau pathology may be differentially associated with Alzheimer's disease (AD)-related demographic, cognitive, genetic and neuroimaging markers. We examined 771 participants with normal cognition, mild cognitive impairment or dementia from BioFINDER-2 ($n = 400$) and ADNI ($n = 371$). All had tau-PET ([18F]RO948 in BioFINDER-2, [18F]flortaucipir in ADNI) and CSF p-tau181 biomarkers available. Plasma p-tau181 and plasma/CSF p-tau217 were available in BioFINDER-2 only. Concordance between PET, CSF and plasma tau biomarkers ranged between 66 and 95%. Across the whole group, ridge regression models showed that increased CSF and plasma p-tau181 and p-tau217 levels were independently of tau PET associated with higher age, and APOEε4-carriership and Aβ-positivity, while increased tau-PET signal in the temporal cortex was associated with worse cognitive performance and reduced cortical thickness. We conclude that biofluid and neuroimaging markers of tau pathology convey partly independent information, with CSF and plasma p-tau181 and p-tau217 levels being more tightly linked with early markers of AD (especially Aβ-pathology), while tau-PET shows the strongest associations with cognitive and neurodegenerative markers of disease progression.

**Keywords** Alzheimer's disease; CSF; PET; plasma; tau
**Subject Categories** Biomarkers; Neuroscience

## Introduction

The core neuropathological features of Alzheimer's disease (AD) are amyloid-β (Aβ) plaques and hyperphosphorylated tau (p-tau) in neuronal neurofibrillary tangles and neuropil threads (Scheltens et al, 2021). While Aβ pathology is often considered a key initiator of AD development (potentially through facilitating the spread of tau pathology, Busche & Hyman, 2020), the phosphorylation, release and aggregation of tau proteins are tightly linked to the clinical and biological progression of AD (Savva et al, 2009; Nelson et al, 2012; Jack & Holtzman, 2013; Spires-Jones & Hyman, 2014; Mattsson-Carlgren et al, 2020a). Major scientific breakthroughs over the past decades now enable the detection of tau pathology in cerebrospinal fluid (CSF), using positron emission tomography (PET) and, most recently, in blood (Ashton et al, 2021a; Bischof et al, 2021; Leuzy et al, 2021; Wolters et al, 2021).

1 Clinical Memory Research Unit, Lund University, Lund, Sweden
2 Alzheimer Center Amsterdam, Department of Neurology, Amsterdam Neuroscience, Vrije Universiteit Amsterdam, Amsterdam UMC, Amsterdam, The Netherlands
3 Department of Health Technologies, Tallinn University of Technology, Tallinn, Estonia
4 Radiology Centre, North Estonia Medical Centre, Tallinn, Estonia
5 Department of Neurology, Skåne University Hospital, Lund, Sweden
6 Memory Clinic, Skåne University Hospital, Malmö, Sweden
7 Clinical Neurochemistry Laboratory, Sahlgrenska University Hospital, Mölndal, Sweden
8 Department of Psychiatry and Neurochemistry, Institute of Neuroscience and Physiology, The Sahlgrenska Academy at the University of Gothenburg, Mölndal, Sweden
9 Department of Neurodegenerative Disease, UCL Institute of Neurology, London, UK
10 UK Dementia Research Institute at UCL, London, UK
11 Eli Lilly and Company, Indianapolis, IN, USA
12 Wallenberg Centre for Molecular Medicine, Lund University, Lund, Sweden
*Corresponding author. Tel: +31 20 44440685; E-mail: r.ossenkoppele@amsterdamumc.nl
**Corresponding author. Tel: +46 072 226 7745; E-mail: oskar.hansson@med.lu.se
†Data used in preparation of this article were obtained from the Alzheimer's Disease Neuroimaging Initiative (ADNI) database (adni.loni.usc.edu). As such, the investigators within the ADNI contributed to the design and implementation of ADNI and/or provided data but did not participate in analysis or writing of this report. A complete listing of ADNI investigators can be found at: http://adni.loni.usc.edu/wp-content/uploads/how_to_apply/ADNI_Acknowledgement_List.pdf

Although these three biomarker modalities are reflecting tau pathology, there are important differences between them. For example, in CSF and plasma-specific soluble variants of tau (e.g. p-tau181 or p-tau217) are measured (Mielke et al, 2018; Barthelemy et al, 2020; Janelidze et al, 2020a; Janelidze et al, 2020b; Karikari et al, 2020; Palmqvist et al, 2020; Thijssen et al, 2020), while tau PET ligands bind aggregated non-soluble paired helical filaments of tau (Xia et al, 2013; Marquie et al, 2015; Hostetler et al, 2016; Kuwabara et al, 2018; Lemoine et al, 2018). PET and fluid biomarkers thus measure different aspects of abnormalities in tau metabolism. Moreover, previous studies have shown that CSF p-tau markers become abnormal prior to tau PET and may thus be more sensitive biomarkers for early AD (Mattsson et al, 2017; Meyer et al, 2020; Reimand et al, 2020b). Similarly, there is emerging evidence that alterations in plasma p-tau levels also occur early in the disease process (Barthelemy et al, 2020; Mattsson-Carlgren et al, 2020a; Mattsson-Carlgren et al, 2020b; Suarez-Calvet et al, 2020; Ashton et al, 2021b; Janelidze et al, 2021; Moscoso et al, 2021). Despite the aforementioned differences among biofluid- and PET-based tau biomarkers, they are often considered interchangeable. For example, CSF p-tau and tau PET are both incorporated as markers of tau pathology in the most recent research criteria for AD (Jack et al, 2018a), and both can be used to define "T" (tau) status in the AT (N) classification system (Jack et al, 2016).

In this study, we investigated whether PET, CSF and plasma biomarkers of tau pathology are differentially associated with AD-related demographic, cognitive, genetic and neuroimaging markers. We hypothesized that the three modalities would show significant differences in associations with distinct AD features and thus partially convey independent information. Exploring this hypothesis might provide insight into the clinical and neurobiological factors related to discrepant results between PET-, CSF- and blood-based biomarkers of tau pathology. Additionally, in light of the recent FDA approval of one of the tau PET tracers (i.e. [18F]flortaucipir, Fleisher et al, 2020), and the rapid development of blood-based p-tau biomarkers, clinicians in specialized care settings may soon have multiple tau biomarker options at their disposal. It is therefore of high clinical relevance to determine the degree of agreement between the tau biomarkers and to identify potential scenarios where one tau biomarker might be preferred over the other(s).

# Results

### Study participants

For this study, we stratified participants by their cognitive status into cognitively unimpaired (CU) and symptomatic groups (combining MCI and dementia, Table 1). As expected in both BioFINDER-2 and ADNI, participants in the symptomatic group were older, had more often pathological levels of both tau and Aβ biomarkers and demonstrated worse cognitive performance and greater atrophy on MRI compared to CU individuals.

### Associations between tau biomarkers and AD-related features

First, we examined the correlations between continuous tau biomarkers and other AD-related features. Overall, there was a moderate to strong positive correlation between PET, CSF and plasma tau biomarkers (range: 0.46–0.98, all $P < 0.001$, Fig 1A). In addition, tau biomarkers were positively correlated with age (range: 0.22 for plasma p-tau217 to 0.35 for plasma p-tau181, all $P < 0.001$) and amyloid PET global SUVR (range: 0.48 for plasma p-tau181 to 0.78 for PET entorhinal cortex, all $P < 0.001$). Furthermore, tau biomarkers were negatively correlated with CSF Aβ$_{42/40}$ ratio (range: −0.45 for plasma p-tau181 to −0.66 for CSF p-tau181, all $P < 0.001$), cognitive tests scores (e.g. MMSE, range: −0.36 for plasma p-tau181 to −0.75 for PET temporal meta-ROI [weighted average of entorhinal, amygdala, parahippocampal, fusiform and inferior and middle temporal cortex], all $P < 0.001$) and MRI measures (e.g. AD-signature cortical thickness [comprising bilateral entorhinal, inferior and middle temporal and fusiform cortex], range: −0.39 for plasma p-tau181 to −0.62 for PET entorhinal cortex, all $P < 0.001$ Fig 1B). In general, the correlation coefficients for AD-related features were slightly lower for plasma p-tau181. Furthermore, p-tau217 in both plasma and CSF showed a higher correlation with tau PET than p-tau181 in plasma or CSF. Similar associations were observed for the available variables in ADNI (Fig 1C).

### Concordance of PET, CSF and plasma tau biomarkers

Next, we examined the concordance between PET, CSF and plasma tau biomarkers in the BioFINDER-2 cohort. The concordance between tau PET standardized uptake value ratios (SUVR) in the temporal meta-ROI and CSF p-tau181 was 83% (Fig 2A). Of the 17% discordant participants, 16% showed isolated tau positivity on CSF p-tau181 (CSF$^+$), while only 1% were positive for tau PET (PET$^+$) and negative for CSF p-tau181. Tau PET was concordant with plasma p-tau181 in 80% (5% plasma$^+$/PET$^-$, 15% PET$^+$/plasma$^-$), with plasma p-tau217 in 86% (12% plasma$^+$/PET$^-$, 2% PET$^+$/plasma$^-$) and with CSF p-tau217 in 80% of cases (19% CSF$^+$/PET$^-$, 1% PET$^+$/CSF, Fig 2B–D).

The concordance between fluid tau biomarkers ranged between 66% (CSF p-tau217 vs plasma p-tau181), 70% (CSF p-tau181 vs plasma p-tau181), 73% (plasma p-tau181 vs plasma p-tau217), 82% (CSF p-tau181 vs plasma p-tau217), 83% (CSF p-tau217 vs plasma p-tau217) and 95% (CSF p-tau181 vs CSF p-tau217, Fig EV1). Tau PET SUVR in the entorhinal cortex showed a concordance of 87% with CSF p-tau217 (12% CSF$^+$/PET$^-$, 1% PET$^+$/CSF$^-$) and of 84% with plasma p-tau217 (9% plasma$^+$/PET$^-$, 6% PET$^+$/plasma$^-$), Fig EV1. Overall, in terms of biomarker discordance, a biofluid$^+$/PET$^-$ profile was more common than a biofluid-/PET$^+$ profile, with the exception of plasma p-tau181.

### Partly differential associations between tau PET and CSF p-tau181 vs AD-related features

Based on three sets of ridge regression models (model 1: CSF p-tau181, model 2: tau PET SUVR in the temporal meta-ROI, model 3: CSF and PET combined) for each AD-related feature, we examined the possible differential associations for either tau PET or CSF p-tau181 to other AD-related features (Fig 3A). Ridge regression is a statistical approach used to estimate (in this study standardized) β-coefficients in multiple regression models where the independent variables are highly correlated. Analyses across the whole group indicated that CSF p-tau181 was independently from tau PET

**Table 1.   Participant characteristics.**

| | BioFINDER-2 | | | ADNI | | |
|---|---|---|---|---|---|---|
| | Total | CU | MCI/Dem | Total | CU | MCI/Dem |
| n | 400 | 219 | 181 | 371 | 242 | 129 |
| Age | 67.7 (11.4) | 64.57 (12.8) | 71.5 (7.8)* | 73.2 (7.7) | 72.7 (7.2) | 74.3 (8.5)* |
| Sex, female (%) | 195 (49) | 104 (47) | 91 (50) | 206 (56) | 149 (62) | 57 (44)* |
| Dementia (%) | 85 (21) | 0 (0) | 85 (47) | 32 (9) | 0 (0) | 32 (25) |
| Education, years | 12.7 (4.0) | 12.8 (3.5) | 12.6 (4.5) | 16.6 (2.4) | 16.9 (2.2) | 16.1 (2.6)* |
| APOE ε4 carriership (%) | 211 (53) | 97 (44) | 114 (63)* | 145 (39) | 88 (36) | 57 (44) |
| Tau biomarkers | | | | | | |
| Plasma p-tau181 | 8.1 (5.58) | 6.3 (4.5) | 10.3 (6.0)* | – | – | – |
| Plasma p-tau217 | 3.0 (3.68) | 1.3 (1.6) | 5.0 (4.4)* | – | – | – |
| CSF p-tau181 | 94.8 (87.4) | 55.6 (39.9) | 142.2 (104.4)* | 24.5 (12.8) | 22.2 (10.7) | 28.9 (15.2)* |
| CSF p-tau217 | 219.9 (281.2) | 88.1 (106.5) | 379.4 (338.9)* | – | – | – |
| Tau PET entorhinal | 1.40 (0.45) | 1.16 (0.19) | 1.69 (0.50)* | 1.21 (0.22) | 1.15 (0.14) | 1.33 (0.28)* |
| Tau PET temporal meta-ROI | 1.42 (0.54) | 1.18 (0.17) | 1.71 (0.67)* | 1.26 (0.26) | 1.20 (0.13) | 1.39 (0.37)* |
| AD-features | | | | | | |
| CSF Aβ$_{42/40}$ | 0.08 (0.03) | 0.09 (0.03) | 0.06 (0.03)* | 0.06 (0.03) | 0.07 (0.02) | 0.05 (0.03)* |
| Amyloid PET SUVR/CL | 0.73 (0.20) | 0.68 (0.15) | 0.84 (0.23)* | 34.3 (43.1) | 25.6 (35.5) | 50.7 (50.8)* |
| MMSE | 26.6 (4.2) | 28.9 (1.2) | 23.7 (4.7)* | 28.2 (2.4) | 29.1 (1.3) | 26.5 (3.1)* |
| Memory composite z-score | −1.14 (1.53) | −0.12 (0.88) | −2.44 (1.14)* | 0.70 (0.78) | 1.05 (0.56) | 0.04 (0.72)* |
| Language composite z-score | −0.91 (1.46) | −0.12 (0.86) | −1.88 (1.47)* | 0.65 (0.90) | 0.97 (0.75) | 0.05 (0.85)* |
| Executive functioning composite z-score | −1.00 (1.36) | −0.18 (0.77) | −2.01 (1.23)* | 0.81 (1.03) | 1.16 (0.81) | 0.16 (1.08)* |
| Visuospatial composite z-score | −0.84 (2.55) | −0.04 (0.72) | −1.88 (3.54)* | 0.03 (0.78) | 0.15 (0.69) | −0.19 (0.90)* |
| MRI Hippocampal volume/TIV ratio | 2.28 (0.37) | 2.45 (0.29) | 2.09 (0.36)* | 2.47 (0.37) | 2.58 (0.31) | 2.27 (0.39)* |
| MRI AD-signature region thickness | 2.61 (0.22) | 2.72 (0.16) | 2.48 (0.21)* | 2.93 (0.22) | 2.99 (0.18) | 2.81 (0.25)* |

Participant characteristics for the discovery cohort (i.e. BioFINDER-2) and the validation cohort (i.e. ADNI) are presented as mean (SD) or n (%). Significant differences ($P < 0.05$) between the cognitively unimpaired (CU) and the symptomatic (including mild cognitive impairment and dementia) groups are marked with an asterisk in the MCI/Dem group column. Several values are not directly comparable between the discovery cohort and validation cohort, including (i) CSF p-tau181 (assays differ), (ii) tau PET (RO948 in BioFINDER-2, flortaucipir in ADNI), (iii) amyloid PET (SUVR for BioFINDER-2, Centiloids for ADNI, (v) cognitive test composite scores (composed of different tests and standardization differed. MRI Hippocampal volume/TIV ratio values are multiplied by 1,000.

associated with risk factors for AD (i.e. age and APOE ε4 carriership) and Aβ pathology, whereas tau PET was more strongly associated with indicators of disease progression including lower cognitive test scores and reduced cortical thickness. When stratifying for cognitive status (i.e. CU vs MCI/dementia), this pattern was similar for CSF in CU and for PET in the symptomatic stages. Additionally, CSF was more strongly associated with MMSE and executive functioning composite score in CU individuals. These patterns of results were consistent when investigating percentual change of β-coefficients (Appendix Table S1) and when comparing the R-squared values between the simple models for PET and CSF (Appendix Table S2). When dichotomizing for amyloid status in both the total cohort and among CU participants, the stronger association of PET with reduced cognitive performance and cortical thickness was evident only in the amyloid-positive groups (Fig EV2). Replication of the analyses in ADNI revealed overall similar results, except that the stronger association of CSF with cognitive measures in CU individuals observed in BioFINDER-2 was not present in ADNI (Fig 3B). Ridge regression models were superior to ordinary least regression models in the ADNI validation cohort (e.g. $R^2 = 0.31$ vs $R^2 = 0.07$ for MMSE in the combined model), confirming that the L2 regularization performed as expected.

### Relative importance of AD features in predicting CSF p-tau181 and tau PET levels

Next, we explored the association between CSF p-tau181 and tau PET SUVR in a temporal meta-ROI with AD-related features using regression tree models in both BioFINDER-2 (Fig 4A) and in ADNI (Fig 4B). Regression tree modelling is an iterative process that splits data into partitions (or branches) and then continues splitting each partition into smaller groups until further splitting is no longer supported by the data. The advantage of regression tree models is that all variables are modelled together, resulting in an estimation of how important each AD-related feature is relative to the other features for PET and CSF separately. We found that amyloid PET retention and the CSF Aβ$_{42/40}$ ratio were among the most important

**A** BioFINDER-2

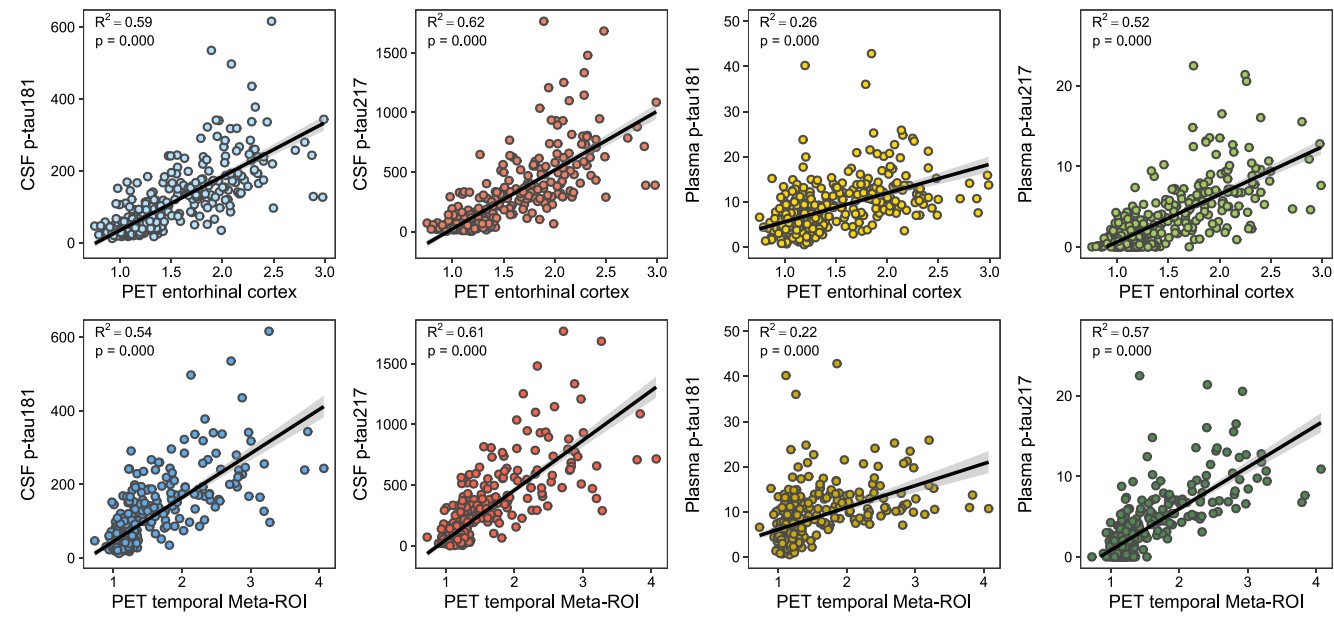

**B** BioFINDER-2

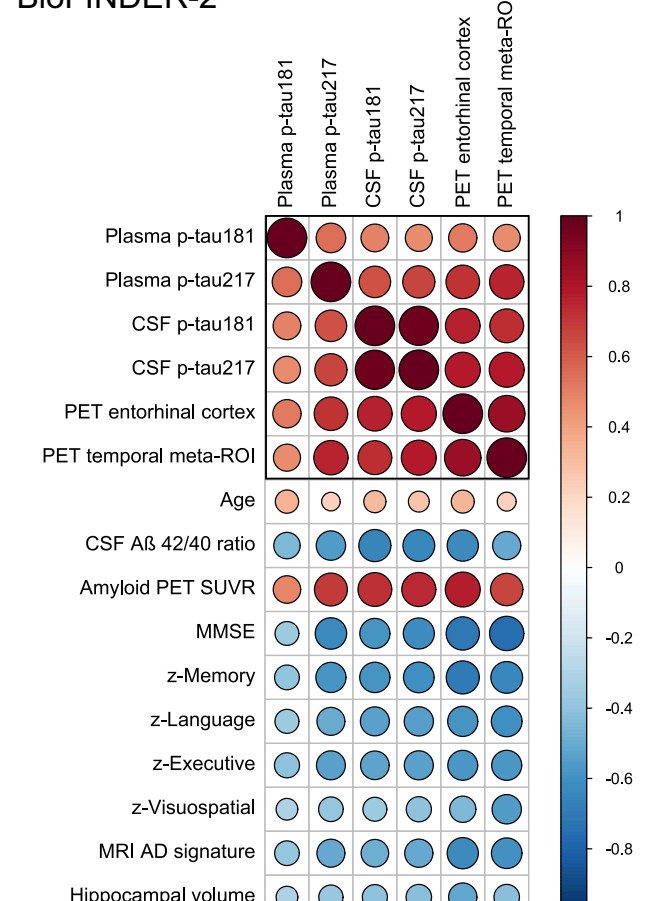

**C** ADNI

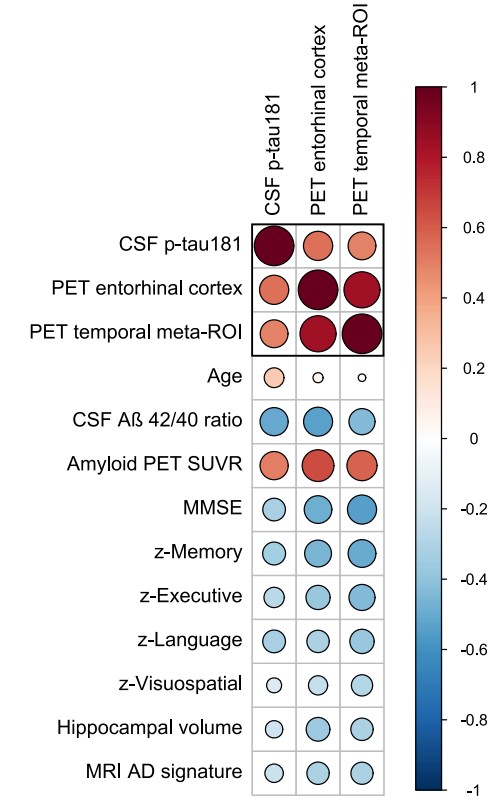

**Figure 1.**

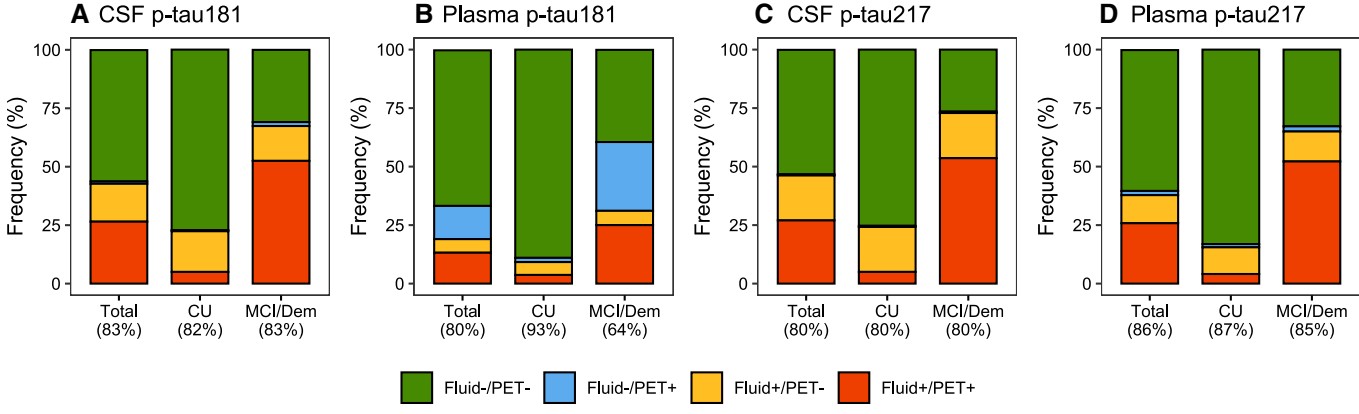

**Figure 1.  Correlations between tau biomarkers and other AD-related features.**

A–C Graphs display the correlations between the tau PET/fluid biomarkers and AD features in BioFINDER-2 (A and B) and ADNI (C). (A) Scatterplots of the associations between tau PET uptake in the entorhinal cortex (upper panel) and in the temporal meta-ROI (lower panel) with from left to right CSF p-tau181, CSF p-tau217, plasma p-tau181 and plasma p-tau217. (B) [BioFINDER-2] and (C) [ADNI] showing matrices of correlation coefficients between all available tau biomarkers and AD-related features. Strong positive correlations are indicated in dark red, while strong negative correlations are indicated in blue (as indicated by Pearson correlation tests).

**Figure 2.  Concordance between tau PET in the temporal meta-ROI and CSF and plasma p-tau181 and p-tau217 in BioFINDER-2.**

A–D The graphs represent concordance rates between the different tau biomarkers in BioFINDER-2. (A) Tau PET in the temporal Meta-ROI vs CSF p-tau181, (B) Tau PET in the temporal meta-ROI vs plasma p-tau181, (C) Tau PET in the temporal meta-ROI vs CSF p-tau217 and (D) Tau PET in the temporal meta-ROI vs plasma p-tau217. Green indicates negative concordance, red indicates positive concordance, blue indicates discordance where tau PET is positive with negative fluid biomarkers, and orange indicates discordance where tau PET is negative with positive fluid biomarkers. Cut-offs for both tau PET and fluid biomarkers are based on the mean + (2 × standard deviation) in Aβ-negative cognitively normal individuals (see Materials and Methods section for further detail).

predictors of tau biomarker levels for most models, but they showed tau biomarker modality specific effects. For example, the CSF Aβ$_{42/40}$ ratio was an important predictor of CSF p-tau181 levels in both CU (median VIM [95% confidence intervals]; BioFINDER-2: 1.23 [1.18–1.32], rank #2; ADNI: 1.35 [1.25–1.51], rank #1) and MCI/dementia (BioFINDER-2: 1.43 [1.32–1.59], rank #1; ADNI: 1.21 [1.10, 1.39], rank #1) groups, while for tau PET it was a more important predictor in the CU group (BioFINDER-2: 0.67 [0.57–0.80], rank #3; ADNI: 0.91 [0.75–1.05], rank #1) than in MCI/dementia (BioFINDER-2: 0.86 [0.72–1.06], rank #9; ADNI: 0.75 [0.67–0.83], rank #4). In CU, age was an important predictor for both CSF p-tau181 and tau PET (ranks varying from #2 to #4), whereas it was not an important predictor for either tau biomarker in the symptomatic stages (ranks varying from #6 to #9). In the symptomatic stages, memory *z*-score was the most informative out of cognitive tests for all models, which was more pronounced for tau PET (ranked #2 in both cohorts) than for CSF p-tau181 (ranked #4 in both cohorts). These results are largely congruent with the information obtained using the ridge regression models.

### Exploring plasma p-tau markers, p-tau217 epitopes and entorhinal cortex tau PET

Finally, we used the aforementioned ridge regression models to explore associations of the AD-related features with plasma p-tau181, plasma p-tau217 and CSF p-tau217 (all compared to temporal meta-ROI tau PET) and with entorhinal cortex tau PET

SUVR (in comparison with CSF p-tau181). When comparing plasma p-tau181 and tau PET in the temporal meta-ROI, plasma p-tau181 only showed a stronger association with age, while tau PET most strongly associated with all cognitive test scores and MRI measures (Fig 5A). Compared to tau PET, plasma p-tau217 was more strongly associated with *APOE* ε4 carriership, amyloid PET and the CSF Aβ$_{42/40}$ ratio, while tau PET was more strongly associated with cognitive decline and MRI-based atrophy compared to plasma p-tau217 (Fig 5B). Replacing CSF p-tau181 with CSF p-tau217 yielded essentially the same results as in our main analysis (Fig 5C), although it should be noted that the R-squared values from the simple models were slightly but consistently higher for CSF p-tau217 compared to CSF p-tau181 Table EV1), with the exception of age. We additionally compared CSF p-tau217 to tau PET in the entorhinal cortex (Fig EV3). Contrary to analyses including the temporal meta-ROI, CSF p-tau217 was no longer independently associated with age and *APOE* ε4 carriership, and the stronger associations between tau PET and cognitive decline in the symptomatic stage were no longer found.

## Discussion

In this study, we aimed to investigate whether PET, CSF and plasma biomarkers of tau pathology in AD are comparable to each other or carry unique information about AD-related demographic, cognitive, genetic and neuroimaging markers. First, we showed that the

## A  Discovery cohort (BioFINDER-2)

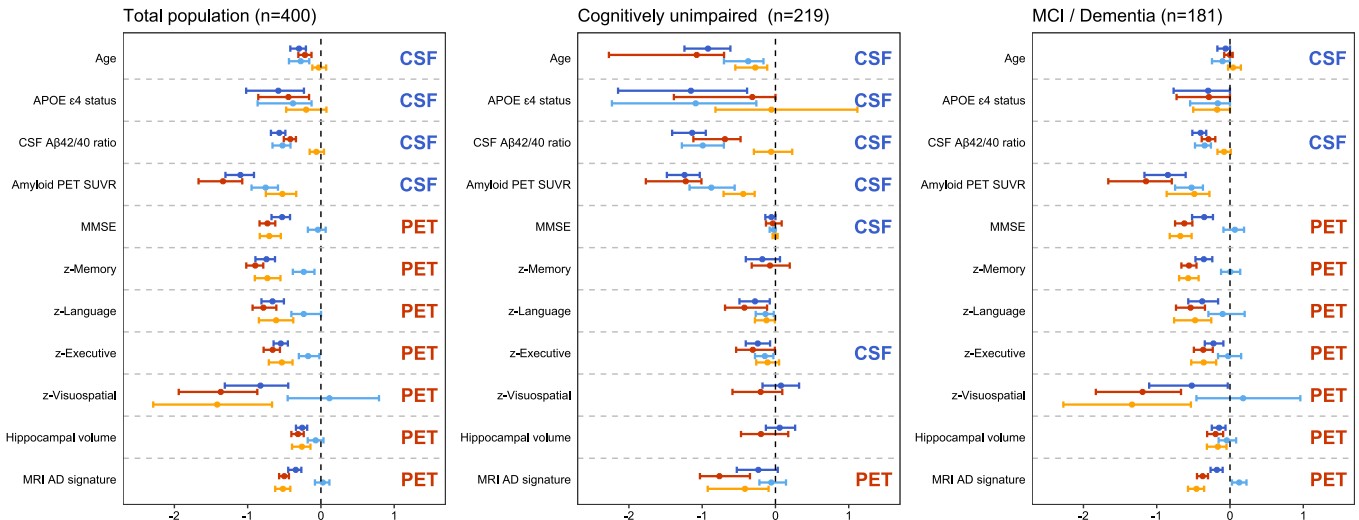

## B  Replication cohort (ADNI)

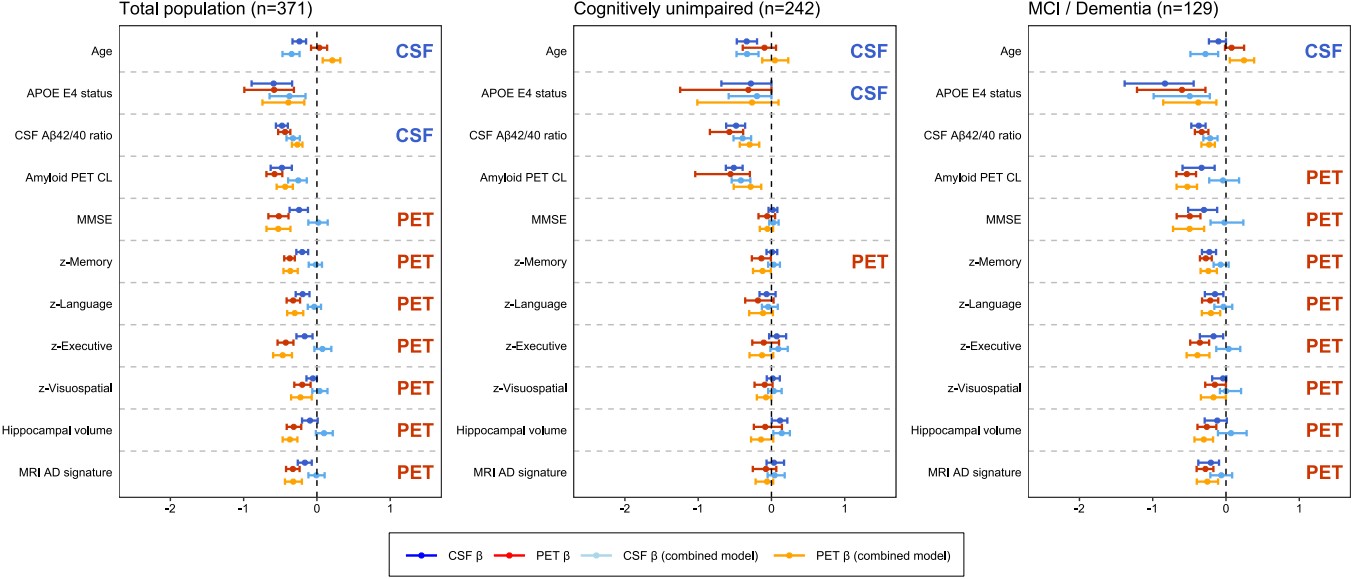

**Figure 3.  Differential associations of tau PET vs CSF p-tau181 with Alzheimer-related features.**

A, B  Graphs display the differential association of temporal Meta-ROI tau PET vs CSF p-tau181 with AD-related features in the discovery cohort BioFINDER-2 (panel A) and the validation cohort ADNI (panel B). Median and 95% confidence intervals (CI) of β-coefficients are plotted from the following ridge regression models: (i) AD-related feature predicted by CSF p-tau181 (dark blue, simple model), (ii) AD-related feature predicted by tau PET (dark red, simple model), (iii) AD-related feature predicted by the combination of CSF p-tau181 (light blue) and tau PET (orange, combined model). In case a feature was non-significant for both tau biomarkers in the simple models (i.e. 95% CIs crossed the 0-line), no combined model was performed. All models were adjusted for age and sex, and cognitive tests were additionally adjusted for education. The β-coefficients for age, *APOE ε4* carriership and amyloid PET global measures were multiplied by −1 for visualization purposes.

To compare the strength of the associations between PET and CSF tau biomarkers with the predicted AD-features, we followed these three criteria: (i) non-overlapping 95% CIs of the β-coefficient of the simple models for CSF and PET (i.e. stronger association for the biomarker with the more positive or negative value), (ii) 95% CIs in the simple or combined models non-overlapping with β = 0 for only CSF or PET (stronger association for the tau biomarker non-overlapping with β = 0), (iii) a significant drop of the β-coefficient from the combined model relative to the simple model for only CSF or PET (i.e. overlapping 95% CIs for one tau biomarker between the simple and combined model, but non-overlapping 95% CIs for the other, with a stronger association for the biomarker with overlapping 95% CIs).

concordance ranged between 66% (CSF p-tau217 vs plasma p-tau181) and 95% (CSF p-tau181 vs CSF p-tau217) across all tau biomarkers, and between 80% (tau PET vs plasma p-tau181/CSF

p-tau217) and 86% (tau PET vs plasma p-tau217) for biofluid- vs neuroimaging-based tau biomarkers. Ridge regression models showed that increased CSF and plasma p-tau181 and p-tau217 levels

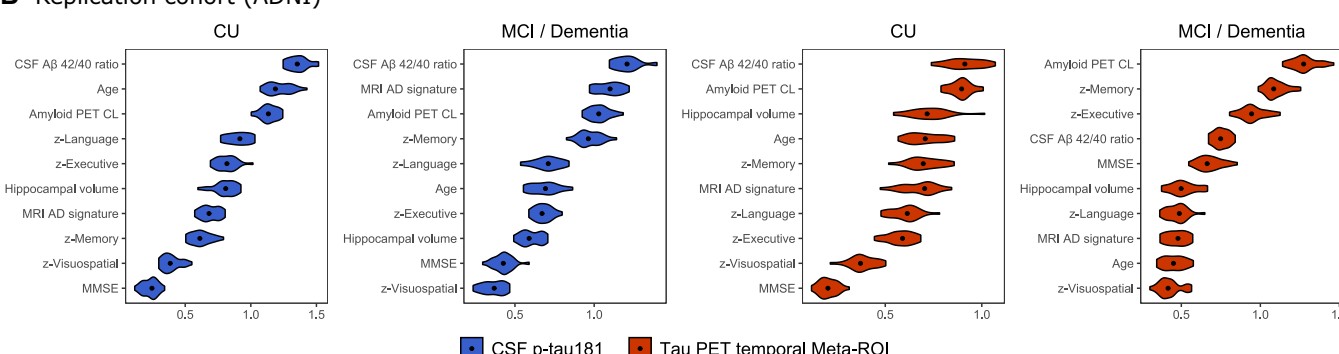

**Figure 4.  Rank-ordering of importance in predicting CSF p-tau181 and tau PET levels.**

A, B  Graphs shows in a rank-ordered manner the most important predictors of CSF p-tau181 (blue) and tau PET temporal meta-ROI (red) levels in BioFINDER-2 (A) and
ADNI (B). The violin plots of the variable importance measures (VIM) are based on the computation of 25 times the regression tree models using multiple
imputation datasets (25×). Median VIM values are marked with a dot.

were independently associated with ageing, and *APOE* ε4 and Aβ positivity, while increased temporal meta-ROI tau PET retention was more strongly associated with worse cognitive performance and reduced cortical thickness. The majority of results were consistent between the discovery cohort (i.e. BioFINDER-2) and the replication cohort (i.e. ADNI). The data suggest that biofluid tau biomarkers are more tightly linked with early markers of AD (especially Aβ pathology), while tau PET showed strongest associations with cognitive and neurodegenerative markers of disease progression. Overall, the results support our hypothesis that the three tau biomarker modalities provide partially independent information.

The concordance between tau PET vs p-tau181 and p-tau217 in CSF and in plasma ranged from 80 to 86%. An optimistic interpretation of this level of agreement would be that CSF and especially plasma biomarkers offer cheaper and more scalable alternatives compared to tau PET when the objective is to obtain evidence of the presence or absence of pathological levels of hyperphosphorylated tau. On the other hand, there is also a substantial mismatch of up to 20% between the neuroimaging vs biofluid markers that can have important ramifications for their application in clinical, investigational and clinical trial settings. One can draw a parallel with studies comparing amyloid PET vs CSF Aβ levels that consistently observed discordance rates of ~ 10–20% (Fagan *et al*, 2006; Landau *et al*, 2013; Palmqvist *et al*, 2016; de Wilde *et al*, 2019). This biomarker discordance was demonstrated to be impactful. For example, persons with abnormal CSF Aβ levels but a normal amyloid PET

scan were more likely to accumulate more Aβ pathology over time and to show faster clinical progression than persons with normal CSF Aβ levels but an abnormal amyloid PET (Palmqvist *et al*, 2017; de Wilde *et al*, 2019; Reimand *et al*, 2020b; Sala *et al*, 2020). Moreover, studies have shown that CSF Aβ42 may yield false-positive results in certain neurological conditions evoking a neuroinflammatory response (Mori *et al*, 2011; Krut *et al*, 2013; Reimand *et al*, 2020a), although this can partially be accounted for by using an Aβ42/40 or Aβ42/p-tau ratio. The intended use of biomarkers is thus highly context dependent, and multiple factors such as patient characteristics, goal of the biomarker assessment, and availability of resources and expertise may weigh in. Although the results of this study are not conclusive, they are in line with previous work highlighting that tau PET might be most useful for the differential diagnosis of dementia (thus late-stage disease) and for tracking disease progression (Jack *et al*, 2018b; Ossenkoppele *et al*, 2018; Pontecorvo *et al*, 2019; Leuzy *et al*, 2020; Pascoal *et al*, 2020; Smith *et al*, 2020), while CSF and plasma tau biomarkers are more sensitive markers that can be used to detect AD in its earliest stages (Mattsson-Carlgren *et al*, 2020b; Meyer *et al*, 2020; Suarez-Calvet *et al*, 2020; Ashton *et al*, 2021b; Janelidze *et al*, 2021). Altogether, these findings challenge the notion that the different tau biomarkers can be used interchangeably (Mattsson-Carlgren *et al*, 2020c).

The independent information provided by PET, CSF and plasma biomarkers may be explained by some inherent biological differences. Autopsy studies have shown that tau PET (at least with

## A  Tau PET vs plasma p-tau181

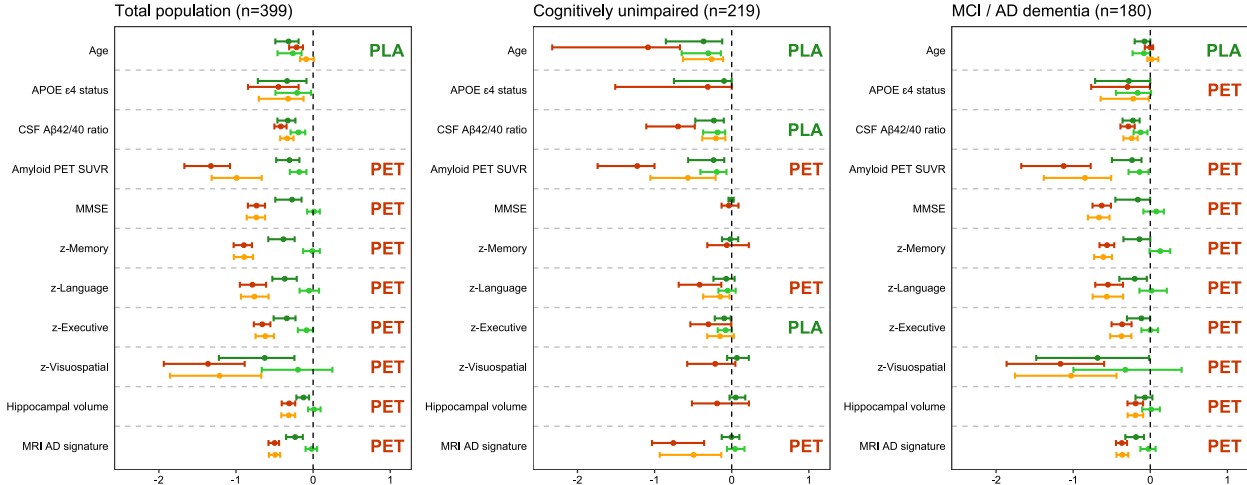

## B  Tau PET vs plasma p-tau217

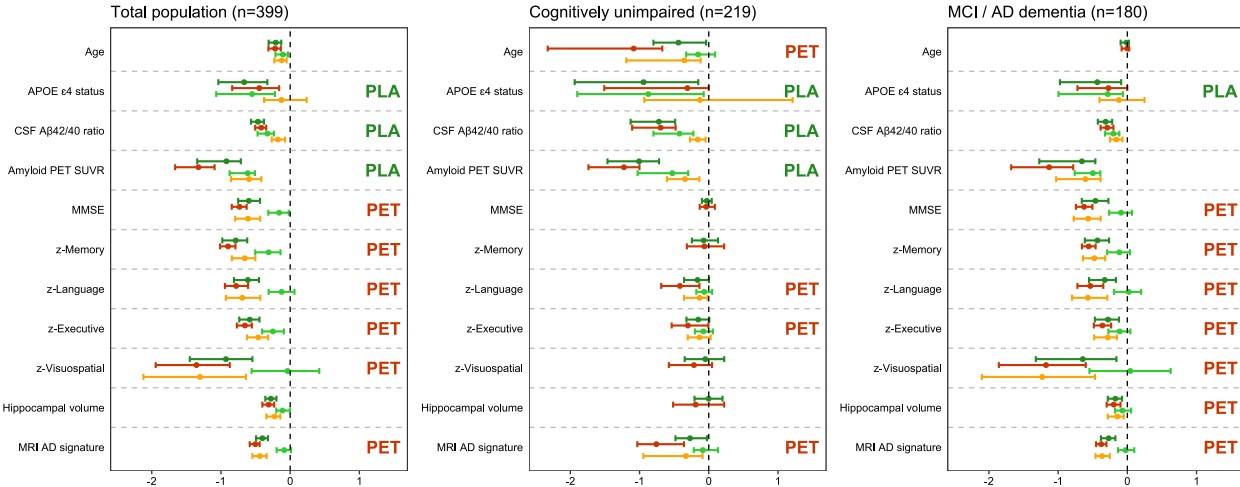

## C  Tau PET vs CSF p-tau217

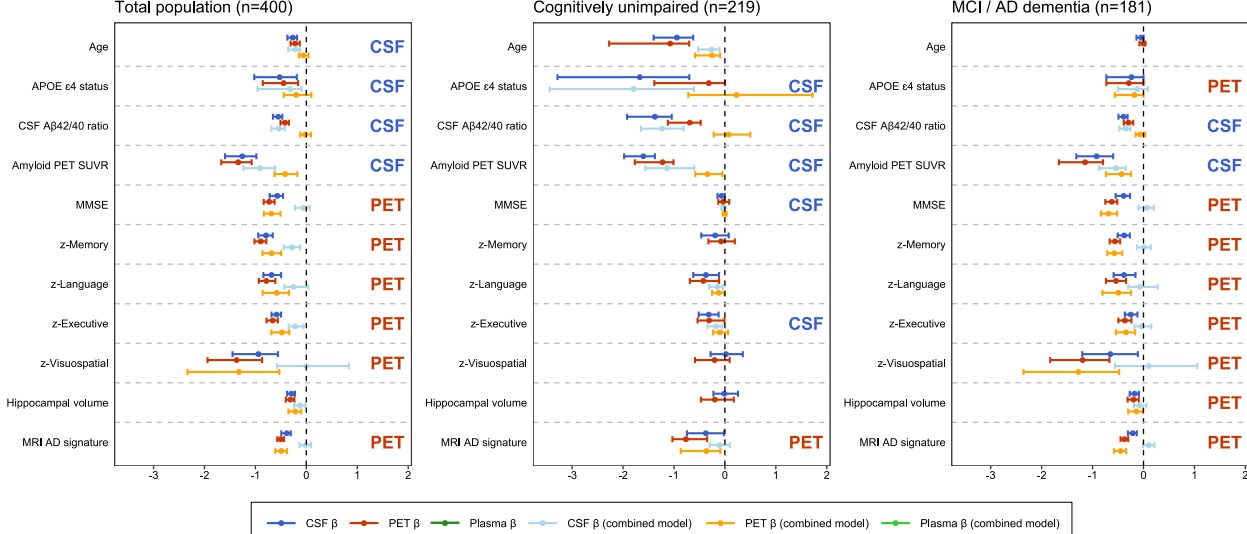

Figure 5.

**Figure 5.  Differential associations of tau PET vs plasma p-tau181, plasma p-tau217 and CSF p-tau217 with Alzheimer related features in BioFINDER-2.**

A–C   Graphs display the differential association of temporal meta-ROI tau PET vs plasma p-tau181 (panel A), vs plasma p-tau217 (panel B) and vs CSF p-tau217 (panel C) with AD-related features in BioFINDER-2. Median and 95% confidence intervals (CI) of β-coefficients are plotted from the following ridge regression models: (i) AD-related feature predicted by fluid biomarkers (dark blue, simple model), (ii) AD-related feature predicted by tau PET (dark red, simple model), (iii) AD-related feature predicted by the combination of fluid biomarkers (light blue) and tau PET (orange, combined model). In case a feature was non-significant for both tau biomarkers in the simple models (i.e. 95% CIs crossed the 0-line), no combined model was performed. All models were adjusted for age and sex, and cognitive tests were additionally adjusted for education. The β-coefficients for age, *APOE* ε4 carriership and amyloid PET global measures were multiplied by −1 for visualization purposes.

To compare the strength of the associations between PET and CSF tau biomarkers with the predicted AD-features, we followed these three criteria: (i) non-overlapping 95% CIs of the β-coefficient of the simple models for CSF and PET (i.e. stronger association for the biomarker with the more positive or negative value), (ii) 95% CIs in the simple or combined models non-overlapping with β = 0 for only CSF or PET (stronger association for the tau biomarker non-overlapping with β = 0), (iii) a significant drop of the β-coefficient from the combined model relative to the simple model for only CSF or PET (i.e. overlapping 95% CIs for one tau biomarker between the simple and combined model, but non-overlapping 95% CIs for the other, with a stronger association for the biomarker with overlapping 95% CIs).

the tracer [$^{18}$F]flortaucipir) is not very sensitive and generally reveals neurofibrillary tangle pathology in Braak stage 4 or higher (Fleisher *et al*, 2020; Lowe *et al*, 2020). On the other hand, CSF and plasma p-tau mirror concurrent abnormalities in tau metabolism such as increased phosphorylation and release of soluble tau from damaged neurons (Jin *et al*, 2011; Mattsson-Carlgren *et al*, 2020a). Soluble p-tau levels in CSF and plasma are thus state markers reflecting the balance between production and clearance of tau at time of lumbar puncture or blood draw, while the insoluble tau aggregates measured with PET are likely the product of processes that have occurred over the entire disease duration, making tau PET a stage marker. The detection of tau pathology in a relatively late stage and in a more mature (insoluble) conformation with PET is in line with its strong cognitive and neurodegenerative correlates (i.e. markers of disease progression), while the detection of early abnormalities in tau metabolism with CSF and plasma is consistent with their strong associations with early indicators of AD-like ageing and *APOE* ε4 and Aβ positivity. In addition to differences between modalities, there were also within-modality differences when using p-tau181 or p-tau217. In line with previous studies (Barthelemy *et al*, 2020; Janelidze *et al*, 2020b), p-tau217 showed subtly stronger associations with tau PET measures and other AD features than p-tau181, which could in turn be explained by stronger relationships with the quantity of neurofibrillary tangle pathology and neuropil threads for antibodies recognizing p-tau217 vs p-tau181 (Spillantini *et al*, 1996).

### Strengths and limitations

Strengths of the study include the large and comprehensive dataset and the inclusion of a replication cohort. There are also some limitations. First, plasma p-tau181 and p-tau217 were not available in ADNI at the time of tau PET (that was performed 6 years later, Moscoso *et al*, 2021) and CSF p-tau217 was not determined. Therefore, we could only replicate the tau PET vs CSF p-tau181 analysis in ADNI. Moreover, we used different tau PET tracers and CSF assays between BioFINDER and ADNI. Second, the sample size of symptomatic patients was insufficient to stratify by MCI and AD dementia and these groups were therefore pooled. Future studies with larger sample sizes should test whether results differ between MCI and AD dementia.

### Future directions

In addition to investigational and research settings, biomarkers have become an integrated part of clinical trials. For example, CSF p-tau

and tau PET have served as a secondary outcome measures in clinical trials testing the efficacy of disease-modifying treatments targeting the Aβ pathway (Salloway *et al*, 2014; Ostrowitzki *et al*, 2017; Sevigny *et al*, 2017). Until recently, (pre-)screening and/or selection of clinical trial participants was only done using Aβ PET and/or CSF biomarkers (Sperling *et al*, 2020). A recent successful phase II clinical trial with the Aβ antibody donanemab, however, took an innovative approach by selecting Aβ-positive individuals MCI/mild dementia with intermediate levels of tau pathology based on a PET scan (Mintun *et al*, 2021). Furthermore, an underpowered exploratory analysis suggested that clinical benefit may be associated with the lower tau PET SUVR range. This clinical trial served as the first example for tau PET biomarkers as a selection tool for trial participants. Future work is needed to establish whether cheaper and more scalable plasma and/or CSF tau biomarkers are suitable alternatives to tau PET. A key question is whether biofluid markers show a greater dynamic range at earlier pathological stages, while PET continues to increase in the advanced disease stages (i.e. plateaus at a later stage of the disease). As clinical trial design continues to evolve, it is important to consider the use of biomarkers in those trials and how those markers may or may not be useful in primary care settings to aid in management of patients through helping inform on when to start or stop treatment.

## Materials and Methods

### Participants

We included a total of 771 participants from the Swedish BioFINDER-2 study at Lund University (discovery cohort, *n* = 400) and Alzheimer's disease neuroimaging initiative (ADNI, replication cohort, *n* = 371), including 461 cognitively unimpaired (CU) individuals (BioFINDER-2: 219, ADNI: 242) and 310 individuals diagnosed with mild cognitive impairment (MCI) or AD dementia according to NIA-AA diagnostic criteria (Albert *et al*, 2011; McKhann *et al*, 2011) (BioFINDER-2: 181, ADNI: 129). All participants had tau PET and CSF data available, while plasma p-tau biomarkers (at time of tau PET and lumbar puncture) were only available for the Swedish BioFINDER-2 participants. In addition, all participants underwent a medical history and neurological examination, MRI, *APOE* genotyping and a neuropsychological test battery that included the Mini-Mental State Examination (MMSE) and domain-specific tests for memory, executive functioning, language

and visuospatial abilities (Crane *et al*, 2017; Ossenkoppele *et al*, 2019; Choi *et al*, 2020). Written informed consent was obtained from all participants and local institutional review boards for human research approved the study. The study was performed in accordance with the ethical standards as laid down in the 1964 Declaration of Helsinki and its later amendments, and the experiments conformed to the principles set out in the Department of Health and Human Services Belmont Report. This study followed the STROBE reporting guidelines.

### MRI data

In BioFINDER-2, a high-resolution T1-weighted MRI was acquired (3T MAGNETOM Prisma; Siemens Healthineers), while multiple 1.5T and 3T MRI scanners were used in the multi-centre ADNI study (Weiner & Veitch, 2015). MRI data were processed using previously reported procedures (Ossenkoppele *et al*, 2018; Ossenkoppele *et al*, 2019; Leuzy *et al*, 2020; Ossenkoppele *et al*, 2020a; Ossenkoppele *et al*, 2020b). Briefly, cortical reconstruction and volumetric segmentation were performed with the FreeSurfer (v6.0) image analysis pipelines (http://surfer.nmr.mgh.harvard.edu/). The MP-RAGE images underwent correction for intensity homogeneity (Sled *et al*, 1998), removal of non-brain tissue (Segonne *et al*, 2004) and segmentation into grey matter (GM) and white matter (WM) with intensity gradient and connectivity among voxels (Fischl *et al*, 2002). Cortical thickness was measured as the distance from the GM/WM boundary to the corresponding pial surface (Fischl & Dale, 2000). Reconstructed datasets were visually inspected for accuracy, and segmentation errors were corrected. We computed hippocampal volumes (adjusted for total intracranial volume) and AD-signature cortical thickness comprising bilateral entorhinal, inferior and middle temporal and fusiform cortex (Jack *et al*, 2017).

### Amyloid PET and CSF

In the BioFINDER-2 study, amyloid PET was performed using [18F] flutemetamol on a digital Discovery MI scanner (GE Healthcare). SUVR images were created for the 90–110 min post-injection interval using the pons as reference region. In ADNI, amyloid PET was performed using [18F]florbetapir ($n = 221$, 50–70 min post-injection, whole cerebellum reference region) or [18F]florbetaben ($n = 150$, 90–110 min post-injection, whole cerebellum reference region) on multiple PET scanners. SUVR values were re-scaled onto the Centiloid scale (for [18F]florbetapir: [196.9 * SUVR] – 196.03, for [18F]florbetaben: [159.08 * SUVR] – 151.65) (Klunk *et al*, 2015) to enable pooled analysis. For BioFINDER-2 and ADNI, the CSF $A\beta_{42/40}$ ratio was determined using the MSD platform (Meso Scale Discovery) and Elecsys immunoassays (Roche Diagnostics, Basel), respectively.

### Tau PET

PET images were processed using previously reported procedures (Maass *et al*, 2017; Ossenkoppele *et al*, 2018; Ossenkoppele *et al*, 2020a; Ossenkoppele *et al*, 2020b). In the BioFINDER-2 study, tau PET was performed using [18F]RO948 on a digital Discovery MI scanner (GE Healthcare). Standardized uptake value ratio (SUVR) images were created for the 70- to 90-min post-injection interval using the inferior cerebellar cortex as the reference region. In ADNI,

tau PET was performed using [18F]flortaucipir and SUVR images were created for the 80–100 post-injection interval using inferior cerebellar cortex as the reference region using a previously published approach (Maass *et al*, 2017). In line with previous work from our group and others (Cho *et al*, 2016; Ossenkoppele *et al*, 2018; Jack *et al*, 2019; Leuzy *et al*, 2020; Ossenkoppele *et al*, 2020a), we used a temporal meta-ROI (Jack *et al*, 2017) comprising a weighted average of entorhinal, amygdala, parahippocampal, fusiform and inferior and middle temporal ROIs for the primary analysis. For the concordance analysis with CSF and plasma p-tau biomarkers, we binarized temporal meta-ROI tau PET retention using previously established cut-offs of 1.36 ([18F]RO948) and 1.34 ([18F]flortaucipir) SUVR based on mean + (2 × standard deviation) uptake in elderly cognitively normal individuals and mean + (2 × standard deviation) uptake in young cognitively normal individuals, respectively (Ossenkoppele *et al*, 2020a). Because previous studies suggested that CSF p-tau and plasma p-tau may become abnormal prior to tau PET, we performed a sensitivity analysis using entorhinal cortex SUVR (Johnson *et al*, 2016; Scholl *et al*, 2016) (a brain region affected early in AD, Braak & Braak, 1991), using previously established cut-offs of 1.48 ([18F]RO948) and 1.39 ([18F]flortaucipir) SUVR (Ossenkoppele *et al*, 2020a).

### CSF p-tau biomarkers

Cerebrospinal fluid samples were derived from lumbar puncture performed within 12 months from the tau PET scan. The procedures and analyses of CSF followed the Alzheimer's Association Flow Chart for CSF biomarkers (Blennow *et al*, 2010) and were performed by technicians blinded to the clinical and imaging data. In BioFINDER-2, analysis of CSF p-tau181 and p-tau217 was performed at Eli Lilly and Company using the MSD platform (Janelidze *et al*, 2020b). In ADNI, CSF p-tau181 was quantified using Elecsys immunoassays (Roche, Basel, Switzerland), while CSF p-tau217 was not available (Lifke *et al*, 2019). For concordance analyses (in BioFINDER-2 only), we used a previously established cut-off for CSF p-tau217 of 101.95 pg/ml, based on the mean + (2 × standard deviation) in a group of 224 Aβ-negative cognitively normal individuals (Palmqvist *et al*, 2020). Because there was no predefined cut-off for CSF p-tau181, we established a cut-off at 69.46 pg/ml based on the mean + (2 × standard deviation) in 200 Aβ-negative cognitively normal individuals from BioFINDER-2 (of whom 111 overlapped with the current sample). Note that cut-offs were only used for the concordance analyses presented in Figs 2 and EV1, with continuous CSF (and plasma) values used in all other statistical models.

### Plasma p-tau biomarkers

Plasma biomarkers were measured as described previously (Palmqvist *et al*, 2020). Plasma p-tau181 was quantified using an in-house Simoa-based immunoassay at the Clinical Neurochemistry Laboratory in Gothenburg (Karikari *et al*, 2020). Analysis of plasma p-tau217 was performed at Eli Lilly and Company using the MSD platform (Mielke *et al*, 2018; Janelidze *et al*, 2020a). For both plasma p-tau181 and p-tau217, one outlier was removed. Out of 399 study participants, 110 had plasma p-tau217 levels below the detection limit of the assay. When plasma p-tau217 concentrations could

not be interpolated from the standard curve due to very low signal, values were imputed to the lowest measurable value. Out of 110 samples below the detection limit, plasma p-tau217 values were imputed for 41 cases (10.3% of the total sample). Note that 92.7% (38/41) of imputed data and 92.7% (102/110) of samples below the detection limit were present in the Aβ-negative group. Therefore, these values were considered to represent truly very low p-tau217 concentrations and were included in all statistical analysis. Cut-offs for concordance analyses were previously established at 11.9 pg/ml for plasma p-tau181 and at 2.5 pg/ml for plasma p-tau217, based on the mean + (2 × standard deviation) in group of 224 Aβ-negative cognitively normal individuals (Palmqvist et al, 2020).

## Statistical analyses

Statistical analysis was performed using R software (Version 4.0.3). When presenting group characteristics, patient features were compared using two samples t-tests and chi-squared tests. Non-adjusted Pearson correlation coefficients were calculated using the *corrplot* package. We accounted for missing values using multiple imputations using the *mice* package (25 imputations and five iterations), as both of the methods described below needed all values to be present. An overview of the proportion of missing values is presented in Appendix Table S3.

First, we examined the correlations between the different tau biomarkers and AD-related features, as well as the degree of concordance between the PET, CSF and plasma tau biomarkers.

Second, we tested whether CSF p-tau181 and tau PET in the temporal meta-ROI were differentially associated with other AD-related features. As those features, we selected age, *APOE* ε4 carriership, amyloid biomarkers (CSF Aβ$_{42/40}$ ratio, amyloid PET global SUVR/Centiloids), cognitive measures (MMSE and composite *z*-scores for memory, language, executive functioning and visuospatial domains (Crane *et al*, 2017; Ossenkoppele *et al*, 2019; Choi *et al*, 2020), and structural MRI measures (hippocampal volumes [adjusted for intracranial volume] and AD-signature cortical thickness). In line with comparable work on amyloid biomarkers (Mattsson *et al*, 2015), we first created three sets of regularized regression models using ridge regression: (i) AD-related feature predicted by CSF p-tau181 (simple model), (ii) AD-related feature predicted by PET temporal meta-ROI (simple model), and (iii) AD-related feature predicted by both CSF p-tau181 and PET temporal meta-ROI (combined model). All models were adjusted for age and sex, and models predicting cognitive performance were additionally adjusted for education. We chose to use ridge regression because it provides stable estimates of β-coefficients despite correlated predictors (Friedman *et al*, 2009), which is the case in our combined models. From this set of analyses, we computed four 95% confidence intervals (CI) of β-coefficients for each AD-related feature (i.e. for PET from the simple model, for CSF from the simple model and for both PET and CSF from the combined model) by using bootstrapped sampling with replacement (N = 1,000 iterations). We used these 95% CIs to compare the β-coefficients of models for PET and CSF. Note that amyloid PET was not performed in cases with dementia in the BioFINDER-2 study. Hence, amyloid PET SUVR values were not imputed (because it would introduce systematic bias), and the bootstrapped samples were taken only from cases with available amyloid PET. To compare the strength of

the associations between PET and CSF tau biomarkers with the predicted AD-features, we followed these three criteria: (i) non-overlapping 95% CIs of the β-coefficient of the simple models for CSF and PET (i.e. stronger association for the biomarker with the more positive or negative value), (ii) 95% CIs in the simple or combined models non-overlapping with β = 0 for only CSF or PET (stronger association for the tau biomarker non-overlapping with β = 0), (iii) a significant drop of the β-coefficient from the combined model relative to the simple model for only CSF or PET (i.e. overlapping 95% CIs for one tau biomarker between the simple and combined model, but non-overlapping 95% CIs for the other, with a stronger association for the biomarker with overlapping 95% CIs). In case a feature was not significantly associated with both tau biomarkers, we did not perform the combined model. Figure EV4 provides a more detailed description of these aforementioned models, including examples of how the outcomes can be interpreted. We also investigated whether the results were similar when dichotomizing for amyloid status based of $^{18}$F-flutemetamol PET global SUVR or CSF Aβ$_{40/42}$ when PET was unavailable, using previously established cut-offs (Mattsson-Carlgren *et al*, 2020c; Palmqvist *et al*, 2020). Additionally, we compared the change in percentage of β-coefficients from the simple to the combined model, and the R-squared values between the simple models of CSF and PET. After the primary analysis in BioFINDER-2, we aimed to replicate the findings using the ADNI dataset. Finally, we assessed whether the L2 regularization in the ridge regression models performed as expected, by comparing R-squared values between ridge regression models and ordinary least regression models in simple and combined models in the discovery cohort (BioFINDER-2) vs the validation cohort (ADNI) for the predictor age, CSF Aβ$_{42/40}$ ratio, amyloid PET, hippocampal volumes and AD-signature cortical thickness.

Third, to identify the AD features that were most strongly associated with the tau biomarkers, we created parallel regression tree models (predicting CSF p-tau181 levels and tau PET SUVR in the temporal meta-ROI) with continuous AD-related patient features using the *caret* package. These regression trees provide an estimate of relative strength of the association between AD-related patient features and tau biomarkers, while adjusting for all other variables in the model (Brieman *et al*, 1984). Regression tree models are based on binary recursive partitioning in which each fork is a split of a predictor variable and each node at the end has a prediction for the continuous outcome variable. The splits are based on minimizing the overall sums of squares error and are pruned down to reduce over-fitting. We used bootstrap aggregation (i.e. bagging) with N = 25 bootstrap replications in which final predictions are based on the average of the replications. Model accuracy is tested using 10-fold cross-validation. We used the variable importance measure (VIM) from the models to estimate the relative importance of variables to predict tau biomarkers. The VIM for a variable is based on the average decrease of root mean squared error when it is used in the model, and higher VIM indicates that the variable is considered more important to predict the outcome. In our analysis, we present the VIMs from N = 25 bagged regression tree models, created with each of the 25 imputed datasets.

Fourth, we aimed to explore p-tau181 in plasma, a different p-tau isoform (i.e. p-tau217) in plasma and CSF, and an earlier affected tau PET region (i.e. entorhinal cortex) in the BioFINDER-2 cohort

**The paper explained**

**Problem**

Major scientific breakthroughs over the past decades now enable the detection of tau pathology in cerebrospinal fluid (CSF), using positron emission tomography (PET) and, most recently, in blood. Although these three biomarker modalities reflect tau pathology and are often considered interchangeable in current diagnostic and research criteria for Alzheimer's disease (AD), there are important differences between them.

**Results**

We examined 771 participants with normal cognition, mild cognitive impairment or dementia from two cohorts: BioFINDER-2 ($n = 400$) and ADNI ($n = 371$). All had tau-PET and CSF p-tau181 biomarkers available, while Plasma p-tau181 and plasma/CSF p-tau217 were available in BioFINDER-2 only. Concordance between PET, CSF and plasma tau biomarkers ranged between 66 and 95%. Furthermore, increased CSF and plasma p-tau181 and p-tau217 levels were independently of tau PET associated with higher age, and $APOE\varepsilon4$-carriership and Aβ-positivity, while increased tau-PET signal in the temporal cortex was associated with worse cognitive performance and reduced cortical thickness.

**Impact**

Our study shows that biofluid and neuroimaging markers of tau pathology convey partly independent information, with CSF and plasma p-tau181 and p-tau217 levels being more tightly linked with early markers of AD (especially Aβ pathology), while tau-PET shows the strongest associations with cognitive and neurodegenerative markers of disease progression.

only. Therefore, we conducted ridge regression models (as described above) to investigate the potentially different associations with AD features using different pairings of tau biomarkers, i.e., (i) tau PET SUVR in the temporal meta-ROI vs plasma p-tau181 levels, (ii) tau PET SUVR in the temporal meta-ROI vs plasma and CSF p-tau217 levels and (iii) tau PET SUVR in the entorhinal cortex vs CSF p-tau181.

## Data availability

Anonymized data will be shared by request from a qualified academic investigator and as long as data transfer is in agreement with EU legislation on the general data protection regulation and decisions by the Ethical Review Board of Sweden and Region Skåne, which should be regulated in a material transfer agreement.

**Expanded View** for this article is available online.

## Acknowledgements

This project has received funding from the European Research Council (949570), the Swedish Research Council (2016-00906), the Knut and Alice Wallenberg foundation (2017-0383), the Marianne and Marcus Wallenberg foundation (2015.0125), the Strategic Research Area MultiPark (Multidisciplinary Research in Parkinson's disease) at Lund University, the Swedish Alzheimer Foundation (AF-939932), the Swedish Brain Foundation (FO2019-0326), The Parkinson foundation of Sweden (1280/20), the Skåne University Hospital Foundation (2020-O000028), Regionalt Forskningsstöd (2020-0314) and the Swedish federal government under the ALF agreement (2018-Projekt0279). The precursor of $^{18}$F-flutemetamol was provided by GE Healthcare in BioFINDER-2 and the precursor of $^{18}$F-RO948 was provided by Roche in BioFINDER-2. HZ is a Wallenberg Scholar. KB is supported by the Swedish Research Council (#2017-00915), the Swedish Alzheimer Foundation (#AF-742881), Hjärnfonden, Sweden (#FO2017-0243) and the Swedish state under the agreement between the Swedish government and the County Councils, the ALF-agreement (#ALFGBG-715986). Data collection and sharing for this project was funded by the Alzheimer's Disease Neuroimaging Initiative (ADNI) (National Institutes of Health Grant U01 AG024904) and DOD ADNI (Department of Defense award number W81XWH-12-2-0012). ADNI is funded by the National Institute on Aging, the National Institute of Biomedical Imaging and Bioengineering, and through generous contributions from the following: AbbVie, Alzheimer's Association; Alzheimer's Drug Discovery Foundation; Araclon Biotech; BioClinica, Inc.; Biogen; Bristol-Myers Squibb Company; CereSpir, Inc.; Cogstate; Eisai Inc.; Elan Pharmaceuticals, Inc.; Eli Lilly and Company; EuroImmun; F. Hoffmann-La Roche Ltd and its affiliated company Genentech, Inc.; Fujirebio; GE Healthcare; IXICO Ltd.; Janssen Alzheimer Immunotherapy Research & Development, LLC.; Johnson & Johnson Pharmaceutical Research & Development LLC.; Lumosity; Lundbeck; Merck & Co., Inc.; Meso Scale Diagnostics, LLC.; NeuroRx Research; Neurotrack Technologies; Novartis Pharmaceuticals Corporation; Pfizer Inc.; Piramal Imaging; Servier; Takeda Pharmaceutical Company; and Transition Therapeutics. The Canadian Institutes of Health Research is providing funds to support ADNI clinical sites in Canada. Private sector contributions are facilitated by the Foundation for the National Institutes of Health (www.fnih.org). The grantee organization is the Northern California Institute for Research and Education, and the study is coordinated by the Alzheimer's Therapeutic Research Institute at the University of Southern California. ADNI data are disseminated by the Laboratory for Neuro Imaging at the University of Southern California.

## Author contributions

RO and JR had full access to all the data in the study and take responsibility for the integrity of the data and the accuracy of the data analysis. Concept and design: RO, JR and OH. Acquisition, analysis or interpretation of data: all authors. Drafting of the manuscript: RO, JR and OH. Critical revision of the manuscript for important intellectual content: RS, AL, OS, SP, ES, HZ, PS, JLD, FB, KB, NM-C and SJ.

## Conflict of interest

HZ has served at scientific advisory boards for Eisai, Denali, Roche Diagnostics, Wave, Samumed, Siemens Healthineers, Pinteon Therapeutics, Nervgen, AZTherapies and CogRx, has given lectures in symposia sponsored by Cellectricon, Fujirebio, Alzecure and Biogen, and is a co-founder of Brain Biomarker Solutions in Gothenburg AB (BBS), which is a part of the GU Ventures Incubator Program. KB has served as a consultant, at advisory boards, or at data monitoring committees for Abcam, Axon, Biogen, JOMDD/Shimadzu. Julius Clinical, Lilly, MagQu, Novartis, Roche Diagnostics and Siemens Healthineers, and is a co-founder of Brain Biomarker Solutions in Gothenburg AB (BBS), which is a part of the GU Ventures Incubator Program. SP has served at scientific advisory boards for Roche and Geras Solutions. JLD is an employee and shareholder of Eli Lilly and Company. OH has acquired research support (for the institution) from AVID Radiopharmaceuticals, Biogen, Eli Lilly, Eisai, GE Healthcare, Pfizer and Roche. In the past 2 years, he has received consultancy/speaker fees from AC Immune, Alzpath, Biogen, Cerveau and Roche. The other authors report no conflicts of interest.

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
