## [Review Process File · EMBO Molecular Medicine]

Tau PET correlates with different Alzheimer's disease related features compared to CSF and plasma p-tau biomarkers

Rik Ossenkoppele, Juhan Reimand, Ruben Smith, Antoine Leuzy, Olof Strandberg, Sebastian Palmqvist, Erik Stomrud, Henrik Zetterberg, Philip Scheltens, Jeffrey Dage, Femke Bouwman, Kaj Blennow, Niklas Mattsson-Carlsson, Shorena Janelidze, and Oskar Hansson

DOI: [10.15252/emmm.202114398](https://doi.org/10.15252/emmm.202114398)

Corresponding author(s): Rik Ossenkoppele (r.ossenkoppele@amsterdamumc.nl)

Review Timeline:

Submission Date:	19th Apr 21
Editorial Decision:	5th May 21
Revision Received:	28th May 21
Editorial Decision:	14th Jun 21
Revision Received:	15th Jun 21
Accepted:	16th Jun 21

Editor: Jingyi Hou

Transaction Report:

Thank you again for submitting your work to EMBO Molecular Medicine. We have now heard back from the three referees who evaluated your manuscript. As you will see from the reports below, the referees acknowledge the potential interest and clinical relevance of the study. However, they also raise a series of concerns about your work, which should be convincingly addressed in a major revision of the present manuscript.

The referees' recommendations are rather clear, and there is no need to reiterate their comments. Importantly, Referee #3 requested additional details and a better description of the method and pointed out that the modeling analyses should be made more accessible to the general audience of EMBO Mol Med. Further, Referee #2 asked for additional analysis in the subgroups within the cohort, which needs to be addressed.

***** Reviewer's comments *****

Referee #1 (Comments on Novelty/Model System for Author):

This is a high-quality study examining the relationship between CSF biomarkers (Ab42/40, p-tau) with TAU pet and cognitive decline.

Referee #1 (Remarks for Author):

Ossenkoppele and coworkers describe the relationship between CSF, PET, and plasma biomarkers in regard to AD tau pathology. The investigators identify high concordance between plasma and CSF levels of different p-tau species as well as p-tau and tau PET. Interestingly fluid p-tau levels were associated with age, amyloid status, and APOE4, while tau PET was linked with cognitive decline and brain atrophy. Thus the authors conclude that fluid p-tau reflect an early phase of AD while tau PET is linked with symptomatic cognitive dysfunction. This is an excellent study and I have only two minor points. 1.) Why do the authors observe a discrepancy in the Ab42/40 ratio between the ADNI and BioFINDER-2 cohorts? Ratios of 0.62-0.92 seem a little high based on other cohorts. Is this consistent with other BioFINDER-2 studies? 2.) In Figure 1B and C, it is a little counterintuitive to have the blue color reflect a correlation coefficient of 1 and red -1. I might suggest switching these colors.

Referee #2 (Comments on Novelty/Model System for Author):

The authors examined 771 participants with normal cognition, mild cognitive impairment or dementia from BioFINDER-2 (n=400) and ADNI (n=371); all had tau-PET ([18F]RO948 in BioFINDER-2, [18F]flortaucipir in ADNI) and CSF p-tau181 biomarkers available; plasma p-tau181 and plasma/CSF p-tau217 were available in BioFINDER-2 only. Across the whole group, regression analysis showed that increased CSF and plasma p-tau181 and p-tau217 levels were independently of tau PET associated with A β -positivity, while increased tau-PET signal in the temporal cortex was associated with worse cognitive performance and reduced cortical thickness. They conclude that biofluid and neuroimaging markers of tau pathology convey partly independent information, with CSF and plasma p-tau181 and p-tau217 levels being more tightly linked with early markers of AD (especially A β pathology), while tau-PET shows the strongest associations with cognitive and neurodegenerative markers of disease progression.

It appears that the control and MCI/dementia group have been defined by their clinical status, regardless of their biomarker status. It appears essential to perform the primary analysis not only within the total cohort, but also in the subgroups defined by clinical status AND ATN-biomarker status, since the parameters and correlations reported might behave completely different in the etiologically defined subgroups.

Referee #2 (Remarks for Author):

The authors examined 771 participants with normal cognition, mild cognitive impairment or dementia from BioFINDER-2 (n=400) and ADNI (n=371); all had tau-PET ([¹⁸F]RO948 in BioFINDER-2, [¹⁸F]flortaucipir in ADNI) and CSF p-tau181 biomarkers available; plasma ptau181 and plasma/CSF p-tau217 were available in BioFINDER-2 only. Across the whole group, regression analysis showed that increased CSF and plasma p-tau181 and p-tau217 levels were independently of tau PET associated with A β -positivity, while increased tau-PET signal in the temporal cortex was associated with worse cognitive performance and reduced cortical thickness. They conclude that biofluid and neuroimaging markers of tau pathology convey partly independent information, with CSF and plasma p-tau181 and p-tau217 levels being more tightly linked with early markers of AD (especially A β pathology), while tau-PET shows the strongest associations with cognitive and neurodegenerative markers of disease progression.

It appears that the control and MCI/dementia group have been defined by their clinical status, regardless of their biomarker status. It appears essential to perform the primary analysis not only within the total cohort, but also in the subgroups defined by clinical status AND ATN-biomarker status, since the parameters and correlations reported might behave completely different in the etiologically defined subgroups.

Referee #3 (Comments on Novelty/Model System for Author):

I see no issues with the analysis performed. My only wish is that the authors would share more details about the models created and their predictive ability.

Referee #3 (Remarks for Author):

EMBO Review

The manuscript titled "Independent information from PET, CSF and plasma biomarkers of tau pathology in Alzheimer's disease" addresses a highly relevant question regarding what stages of AD correlate with changes in tau-PET and tau fluid biomarkers. In this study, the authors use sophisticated statistical methods to investigate which AD-related features are best at predicting either tau-PET or tau fluid biomarkers. The analysis is used to investigate what AD-related features correlate best with either tau-PET and tau fluid biomarkers. The study shows that tau fluid biomarkers are more closely correlated with early AD related features such as CSF A β 42/40 ratio and Amyloid PET, whereas tau-PET is more closely related to later AD-related features such as cognitive decline and volumetric MRI changes. I have a few suggestions for changes listed below, but overall the paper is highly relevant and while I am not an expert on the statistical methods used, the analysis appear to be well thought out.

Major comments:

The title does not read well, there seems to be a missing comma and it seems that the intention is for it to read "independent information of X and Y in Alzheimer's disease" - but I am not sure what "independent information" means in this context. What the paper shows is that "Tau-PET correlate with different AD related features than CSF and plasma biomarkers of tau pathology"

The paper relies heavily on statistical modelling - specifically ridge regression and regression trees. These two methods may need to be introduced to the non-statistician reader a little more gently. A couple of sentences describing what these two methods do and how they are interpreted would go

a long way towards bringing readers up to speed with these methods.

In addition, it would be very useful if the actual models and implementation of ridge regression equations could be outlined in the supplemental information. This would allow the non-technical reader to get a better idea of how the regression was implemented. The only reference provided in the paper was to what appears to be a statistical textbook. It is unclear if the ridge regression used here is L2 regularization or some other method. For example, it is unclear if the models in fig 3 are taking all the features into account, or if the models focus on one feature at the time. Similarly, the description of the combined vs individual models in figure 3 does not fully allow this reader to intuit how the equations were actually setup.

For the models, the actual performance of the final models seems to be a relevant result that is missing from the paper. If the ridge regression performed is L2 regularization - then the performance of the original and optimized models would be of interest to the reader as it would allow the reader to assess the effect of the regularization - and the performance of the two models could be tested in the external (ADNI) cohort to ensure that the regularization leads to better performance in new data sets as expected when regularization is performed. Not looking for extensive discussion on this topic as that is not the purpose of the paper - but a couple of sentences regarding performance would be relevant to let the reader know that the regularization did perform as expected.

Minor points:

PET temporal meta-ROI - this term is not defined in the paper.

Page 7 middle of first paragraph states: "while it was a more important predictor of tau PET in the CU group" - when describing Figure 4. However, looking at the figure it seems that the word "more" should have been "less"?

REFEREE #1 (Comments on Novelty/Model System for Author):

This is a high-quality study examining the relationship between CSF biomarkers (Ab42/40, p-tau) with TAU pet and cognitive decline.

Authors' reply: We thank the reviewer for the positive evaluation of our manuscript.

REFEREE #1 (Remarks for Author):

Ossenkoppele and coworkers describe the relationship between CSF, PET, and plasma biomarkers in regard to AD tau pathology. The investigators identify high concocordance between plasma and CSF levels of different p-tau species as well as p-tau and tau PET. Interestingly fluid p-tau levels were associated with age, amyloid status, and APOE4, while tau PET was linked with cognitive decline and brain atrophy. Thus the authors conclude that fluid p-tau reflect an early phase of AD while tau PET is linked with symptomatic cognitive dysfunction. This is an excellent study and I have only two minor points. 1.) Why do the authors observe a discrepancy in the Ab42/40 ratio between the ADNI and BioFINDER-2 cohorts? Ratios of 0.62-0.92 seem a little high based on other cohorts. Is this consistent with other BioFINDER-2 studies? 2.) In Figure 1B and C, it is a little counterintuitive to have the blue color reflect a correlation coefficient of 1 and red -1. I might suggest switching these colors.

Authors' reply: We thank the reviewer again for the positive evaluation of our manuscript.

Regarding minor point #1:

Thanks for spotting this, they were multiplied x10 for BioFINDER and not for ADNI. We have adjusted the values accordingly in Table 1.

Regarding minor point #2:

We fully agree with the reviewer and have switched the colors (see Figure 1).

REFEREE #2 (Remarks for Author):

The authors examined 771 participants with normal cognition, mild cognitive impairment or dementia from BioFINDER-2 (n=400) and ADNI (n=371); all had tau-PET ([18F]RO948 in BioFINDER-2, [18F]flortaucipir in ADNI) and CSF p-tau181 biomarkers available; plasma ptau181 and plasma/CSF p-tau217 were available in BioFINDER-2 only. Across the whole group, regression analysis showed that increased CSF and plasma p-tau181 and p-tau217 levels were independently of tau PET associated with A β -positivity, while increased tau-PET signal in the temporal cortex was associated with worse cognitive performance and reduced cortical thickness. They conclude that biofluid and neuroimaging markers of tau pathology convey partly independent information, with CSF and plasma p-tau181 and p-tau217 levels being more tightly linked with early markers of AD (especially A β pathology), while tau-PET shows the strongest associations with cognitive and neurodegenerative markers of disease progression.

It appears that the control and MCI/dementia group have been defined by their clinical status, regardless of their biomarker status. It appears essential to perform the primary analysis not only within the total cohort, but also in the subgroups defined by clinical status

AND ATN-biomarker status, since the parameters and correlations reported might behave completely different in the etiologically defined subgroups.

In response to this thoughtful comment by the reviewer, we additionally performed our main analysis between tau PET temporal meta-ROI and CSF p-tau181 separately in amyloid-positive and amyloid-negative participants both in the total BioFINDER-2 sample and among CU participants only. The results of this analysis indicate that the higher association between tau PET and lower cognitive test score and MRI atrophy is largely driven by the amyloid-positive participants in the sample. We have added this analysis to the manuscript (new **Figure EV2**) and added the results on page 6.

REFEREE #3 (Comments on Novelty/Model System for Author):

I see no issues with the analysis performed. My only wish is that the authors would share more details about the models created and their predictive ability.

Authors' reply: We thanks the reviewer for the thorough evaluation of our manuscript and have addressed all comments below.

REFEREE #3 (Remarks for Author):

The manuscript titled "Independent information from PET, CSF and plasma biomarkers of tau pathology in Alzheimer's disease" addresses a highly relevant question regarding what stages of AD correlate with changes in tau-PET and tau fluid biomarkers. In this study, the authors use sophisticated statistical methods to investigate which AD-related features are best at predicting either tau-PET or tau fluid biomarkers. The analysis is used to investigate what AD-related features correlate best with either tau-PET and tau fluid biomarkers. The study shows that tau fluid biomarkers are more closely correlated with early AD related features such as CSF Aβ42/40 ratio and Amyloid PET, whereas tau-PET is more closely related to later AD-related features such as cognitive decline and volumetric MRI changes. I have a few suggestions for changes listed below, but overall the paper is highly relevant and

while I am not an expert on the statistical methods used, the analysis appear to be well thought out.

Major comments:

1. The title does not read well, there seems to be a missing comma and it seems that the intention is for it to read "independent information of X and Y in Alzheimer's disease" - but I am not sure what "independent information" means in this context. What the paper shows is that "Tau-PET correlate with different AD related features than CSF and plasma biomarkers of tau pathology"

Authors' reply: We thank the reviewer for this comment. We liked the suggested title so much that we have adopted it.

The paper relies heavily on statistical modelling - specifically ridge regression and regression trees. These two methods may need to be introduced to the non-statistician reader a little more gently. A couple of sentences describing what these two methods do and how they are interpreted would go a long way towards bringing readers up to speed with these methods.

Authors' reply: Thanks for this suggestion. We have shortly explained these methods on page 6 (ridge regression) and page 7 (regression tree modeling).

In addition, it would be very useful if the actual models and implementation of ridge regression equations could be outlined in the supplemental information. This would allow the non-technical reader to get a better idea of how the regression was implemented. The only reference provided in the paper was to what appears to be a statistical textbook. It is unclear if the ridge regression used here is L2 regularization or some other method. For example, it is unclear if the models in fig 3 are taking all the features into account, or if the models focus on one feature at the time. Similarly, the description of the combined vs individual models in figure 3 does not fully allow this reader to intuit how the equations were actually setup.

Authors' reply: We agree with the reviewer that the reader might benefit from a more thorough description on the methods. The ridge regression (RR) models presented in Fig 3 and Fig 5 were performed using L2 regularization and they only include one variable of interest at a time (e.g., cognitive or imaging markers), while adjusting for age and sex, and additionally education in the case of cognitive tests. In response to this comment by the reviewer, we added a figure in the supplementary material (**Figure EV4**) that includes a more thorough explanation of our procedure and examples of the criteria we used to interpret the ridge regression models. We hope that this will be helpful to the reader and enhance the readability of our paper.

For the models, the actual performance of the final models seems to be a relevant result that is missing from the paper. If the ridge regression performed is L2 regularization - then the performance of the original and optimized models would be of interest to the reader as it would allow the reader to assess the effect of the regularization - and the performance of the two models could be tested in the external (ADNI) cohort to ensure that the regularization leads to better performance in new data sets as expected when regularization is performed. Not looking for extensive discussion on this topic as that is not the purpose of the paper - but

a couple of sentences regarding performance would be relevant to let the reader know that the regularization did perform as expected.

Authors' reply: In response to this thoughtful comment by the reviewer, we investigated the performance of our ridge regression models in the main analysis of tau PET temporal Meta-ROI vs CSF p-tau181 in the total sample from BioFINDER-2. Using single imputation, we created ridge regression models identical as in our main analysis, and then created analogous ordinary least squares models. Then we tested these models using ADNI data. We used scaling because some variables were not directly comparable between ADNI and BioFINDER-2 (e.g., different CSF assays, and amyloid PET SUVR vs Centiloid). As the cognitive test composite scores were already differently scaled, we were unable to include them into this analysis. We investigated the performance of these models by calculating R-squared values. When comparing the ridge regression and ordinary least squares models in BioFINDER-2, the R-squared values were in general similar, although the R-squared values from ordinary least squares models were consistently slightly higher, as expected. When using these models in ADNI data, ridge regression models had consistently slightly higher R-squared values (see Table below), which was especially true in combined models. To conclude, our data indeed shows that the ridge regression models are viable and show a better performance in an external dataset compared to ordinary least square models. Based on this comment of the reviewer, we have added a couple of sentences to confirm that the regularization did perform as expected (see page 7 in the Results and page 18 in the Methods).

	Simple model (PET)				Simple model (CSF)				Combined model			
	Biofinder-2		Validation in ADNI		Biofinder-2		Validation in ADNI		Biofinder-2		Validation in ADNI	
	RR	OLS	RR	OLS	RR	OLS	RR	OLS	RR	OLS	RR	OLS
Age	0.057 7	0.058 0	0.092 7	0.105 3	0.102 3	0.102 7	0.033 9	0.029 8	0.101 6	0.130 4	0.028 6	0.029 8
CSF A $\beta_{42/40}$	0.364 0	0.364 0	0.220 6	0.220 4	0.494 6	0.494 7	0.224 9	0.221 8	0.495 0	0.495 6	0.257 4	0.221 8
Amyloid PET SUVR / CL	0.547 1	0.547 3	0.112 6	0.288 3	0.558 2	0.558 2	0.143 0	0.033 5	0.616 0	0.617 6	0.140 0	0.033 5
MMSE	0.589 6	0.589 8	0.309 3	0.304 1	0.352 8	0.353 1	0.082 6	0.074 2	0.590 0	0.590 2	0.310 5	0.074 2
MRI Hippocampal volume	0.420 0	0.420 1	0.211 6	0.210 2	0.383 5	0.383 8	0.086 2	0.081 0	0.419 0	0.421 3	0.213 2	0.081 0
MRI AD signature	0.526 9	0.527 0	0.169 7	0.166 9	0.392 0	0.392 2	0.048 5	0.043 4	0.527 9	0.527 9	0.166 2	0.043 4

Minor points:

PET temporal meta-ROI - this term is not defined in the paper.

Authors' reply: This was initially only described in the Methods section. Based on this comment, we have additionally defined the PET-based temporal meta-ROI in the results section (page 5) and did the same for the MRI-based AD cortical signature (page 5).

Page 7 middle of first paragraph states: "while it was a more important predictor of tau PET

in the CU group" - when describing Figure 4. However, looking at the figure it seems that the word "more" should have been "less"?

Authors' reply: We understand the confusion. The statement "more" referred to the more prominent role in CU compared to MCI/dementia. We have now rephrased the sentence to make this clearer (page 7).

Thank you for the submission of your revised manuscript to EMBO Molecular Medicine. We have now received the enclosed report from the three referees who were asked to re-assess it. As you will see the referees are now supportive and I am pleased to inform you that we will be able to accept your manuscript pending the following amendments:

1. In the main manuscript file, please do the following.

***** Reviewer's comments *****

Referee #1 (Remarks for Author):

The authors have addressed my points.

Referee #2 (Comments on Novelty/Model System for Author):

the manuscript presents novel valuable information regarding information provided by biomarkers on neurobiology of AD

Referee #2 (Remarks for Author):

all raised issues have been resolved

Referee #3 (Comments on Novelty/Model System for Author):

No change to this answer since previous version of the review.

Referee #3 (Remarks for Author):

Thank you to the authors for the revisions to the manuscript. All my questions have been adequately answered and I greatly appreciate the additional details added to the paper. As stated previously - this is a highly relevant and important study.

The authors have made all requested editorial changes.

We are pleased to inform you that your manuscript is accepted for publication and is now being sent to our publisher to be included in the next available issue of EMBO Molecular Medicine.

Corresponding Author Name: Rik Ossenkoppele & Oskar Hansson

Manuscript Number: EMM-2021-14398